# First Synthesis of DBU-Conjugated Cationic Carbohydrate Derivatives and Investigation of Their Antibacterial and Antifungal Activity

**DOI:** 10.3390/ijms24043550

**Published:** 2023-02-10

**Authors:** Fruzsina Demeter, Patrik Török, Alexandra Kiss, Richárd Kovásznai-Oláh, Zsuzsa Máthéné Szigeti, Viktória Baksa, Fruzsina Kovács, Noémi Balla, Ferenc Fenyvesi, Judit Váradi, Anikó Borbás, Mihály Herczeg

**Affiliations:** 1Department of Pharmaceutical Chemistry, Faculty of Pharmacy, University of Debrecen, Egyetem tér 1, H-4032 Debrecen, Hungary; 2Institute of Biotechnology, Faculty of Science and Technology, University of Debrecen, Egyetem tér 1, H-4032 Debrecen, Hungary; 3Department of Medical Microbiology, Faculty of Medicine, University of Debrecen, H-4032 Debrecen, Hungary; 4Doctoral School of Pharmaceutical Sciences, University of Debrecen, H-4032 Debrecen, Hungary; 5Department of Pharmaceutical Technology, Faculty of Pharmacy, University of Debrecen, Nagyerdei Körút 98, H-4032 Debrecen, Hungary

**Keywords:** cationic carbohydrate, amidinium salt, quaternary ammonium compound, antifungal, antibacterial

## Abstract

The emergence of drug-resistant bacteria and fungi represents a serious health problem worldwide. It has long been known that cationic compounds can inhibit the growth of bacteria and fungi by disrupting the cell membrane. The advantage of using such cationic compounds is that the microorganisms would not become resistant to cationic agents, since this type of adaptation would mean significantly altering the structure of their cell walls. We designed novel, DBU (1,8-diazabicyclo[5.4.0]undec-7-ene)-derived amidinium salts of carbohydrates, which may be suitable for disturbing the cell walls of bacteria and fungi due to their quaternary ammonium moiety. A series of saccharide-DBU conjugates were prepared from 6-iodo derivatives of d-glucose, d-mannose, d-altrose and d-allose by nucleophilic substitution reactions. We optimized the synthesis of a d-glucose derivative, and studied the protecting group free synthesis of the glucose-DBU conjugates. The effect of the obtained quaternary amidinium salts against *Escherichia coli* and *Staphylococcus aureus* bacterial strains and *Candida albicans* yeast was investigated, and the impact of the used protecting groups and the sugar configuration on the antimicrobial activity was analyzed. Some of the novel sugar quaternary ammonium compounds with lipophilic aromatic groups (benzyl and 2-napthylmethyl) showed particularly good antifungal and antibacterial activity.

## 1. Introduction

In our rapidly evolving and changing world, in addition to viral infections, another serious problem is the resistance of bacteria to the currently used antibiotics, which means that these preparations must be continuously developed [1,2,3,4,5]. A less mentioned but at least as serious problem is that human pathogenic fungi can similarly become resistant to the antifungal agents used today [6].

The problem with the current treatment of bacterial and fungal diseases is that most of the drugs used affect the life cycle of microbes [7]. The pathogenic organism can adapt relatively easily to this type of intervention through spontaneous mutation [8]. Diseases caused by fungi can be divided into two large groups: (a) superficial mycosis (infection of skin, hair, nails, genitals), (b) systemic mycosis (fungal diseases of internal organs) [9,10]. In terms of their chemical structure, the antifungal agents can be imidazole ring compounds, derivatives containing a triazole group or antimycotics with a special chemical structure. The first group includes clotrimazole (Figure 1) [11,12], which is a broad-spectrum drug. By inhibiting the biosynthesis of ergosterol, it changes the membrane composition of the fungal cell wall, which leads to permeability disturbances and then cell lysis. Fluconazole (Figure 1) is an antifungal medicine containing a triazole group that can be used orally, intravenously, and vaginally [13,14]. The disadvantage is that it also has a liver-damaging effect. Among the active substances with a special structure is flucytosine, [15,16] which can be administered both orally and intravenously. However, this compound can also have serious side effects such as bone marrow suppression, loss of appetite, diarrhea, vomiting, and psychosis. Terbinafine is an allylamine derivative with highly lipophilic properties, which also inhibits the synthesis of ergosterol by inhibiting the enzyme squalene epoxidase, which plays an important role in the synthesis of the fungal cell wall [17,18]. Amphotericin B, isolated in 1955 from a culture of *Streptomyces nodosus*, is an antifungal polyene-macrolide antibiotic [19,20]. It is used for systemic infections, although it has many side effects. As can be seen from the examples so far, there are many effective agents for the treatment of fungal infections; however, the use of most of the products can have many side effects, i.e., the active ingredients damage human cells as well as microbial cells.

Positively charged structures can also have antibacterial and antifungal effects. Cationic ammonium surfactants, for example trimeric surfactant DTAD (tri(dodecyldimethylammonioacetoxy)diethyltriamine trichloride), excellently inhibits the growth of Gram-negative bacteria and fungi, but has no effect on mammalian cells [21]. Furthermore, there exist carbohydrate derivatives conjugated with DABCO (1,4-diazabicyclo[2.2.2]octane), which also have both antifungal and antibacterial effects [22,23]. A general characteristic of these positively charged compounds is that they exert their effect by destroying the cell wall of fungi and bacteria and inhibiting synthesis [24,25,26]. As a result, resistance does not develop against them, or only with difficulty, and since they do not affect the life cycle of the microorganism, they can have a much longer effect. The presented examples clearly show that the cationic nature of compounds is always ensured by one or more quaternary nitrogen atoms, which is decisive in terms of biological activities.

During our previous work, when we studied the elimination reaction of 6-iodo-hexopyranoside derivatives, completely new 1,8-diazabicyclo(5.4.0)undec-7-ene (DBU)-conjugated quaternary amidinium salts were isolated as side products (I–V, Figure 2) [27,28].

These hitherto unknown DBU-carbohydrate conjugates show a high structural similarity to DABCO-sugar conjugates, which have been proven to have antifungal and antibacterial effects [22,23]. We therefore hypothesized that these new compounds also have antimicrobial effects, and we decided to test the antifungal and antibacterial activity of compounds I–V and also investigate the effect of sugar configuration and protecting group pattern on antimicrobial activity.

## 2. Results and Discussion

### 2.1. Synthesis of the DBU-Conjugated Derivatives

During our recent synthesis of l-hexoses based on C-5 epimerization of d-hexoses (d→l conversion), we used 5,6-unsaturated hexopyranosides as the key intermediates [27,28] These 5,6-unsaturated carbohydrates were prepared from the 6-iodo-hexopyranoside derivatives using different bases (Figure 1). When the amidine base DBU was used as a reagent, in addition to the desired elimination reaction, nucleophilic substitution reaction also took place to some extent due to the nucleophilic behavior of DBU [29,30,31,32,33,34]. We investigated DBU-mediated elimination reactions of variously protected carbohydrates including d-*gluco*, d-*manno*, d-*allo* and d-*altro* configured derivatives [27,28]. We found that the elimination/substitution ratio strongly depended on the sugar configuration and, in the case of *gluco* derivatives, also on the protecting group pattern.

In the case of d-*manno* and d-*altro* configurations, the amidinium salts were formed with moderate yields (**21**, **22**, **23**, **24**; **37**, **38,** 28–42%), while in the case of the d-*allo* configured saccharides, the nucleophilic substitution products (**29**, **30**) were obtained in particularly high (63–76%) yields. In the case of the d-*allo* derivative (**31**), using DABCO as the reagent, the quaternary ammonium salt (**32**) was formed exclusively (yield 86%), which could be explained by the greater nucleophilicity of DABCO relative to DBU.

The fully ether-protected d-glucoside **4** gave the DBU conjugate in an acceptable yield, but in the presence of ester protecting groups (**1**, **5**, **6**), the yield of the amidinium salts was very low. We therefore attempted to increase the yield of the DBU-sugar conjugate **3** by shifting the DBU-mediated reaction of the ester-protected derivative **1** [28] towards nucleophilic substitution (Figure 2). The optimization experiments are presented in Table 1. In our previous work, the reactions were carried out in tetrahydrofuran (THF), which is a preferred solvent for elimination [28]. During the optimization, we first replaced THF with acetonitrile, which, according to the literature, is a favorable solvent for nucleophilic substitution instead of elimination [22,23].

Performing the reaction at room temperature in the presence of 1 equiv. of DBU (entry 2.), the reaction was very sluggish, such that even after two weeks, the reaction mixture still contained traces of the starting material. The main product was the 5,6-unsaturated derivative (**2**, [28] 69%), and the expected amidinium salt **3 [28]** was formed only with a very low yield (8%). By increasing the amount of DBU (1.5 eq., entry 2.), the reaction time was reduced to 4 days and the yield of the expected product slightly increased (**3**, 14%).

We tried to decrease the basicity of DBU by adding glacial acetic acid or *p*-toluene sulfonic acid (*p*-TSA) to the reaction mixture. Using 3 equiv. of DBU and 3 equiv. of glacial acetic acid (entry 3.), the reaction ended with a surprising result. The 6-OAc derivative **39** was isolated with a particularly good yield (76%), the 5,6-unsaturated compound (**2**) was formed with a low yield (10%), and the expected DBU-conjugated derivative was not observed. Using *p*-TSA as the acid, no reaction was observed (entry 4.). Increasing the amount of the reagents to 6 equiv. (entry 5.), the 5,6-unsaturated derivative (**2**) was isolated from the reaction mixture as the main product, and the expected product (**3**) was also obtained with a yield of 18%. Using DBU as a solvent (entry 6.), many degradation side-products were formed and the expected compound (**3**) could only be obtained with a low yield. The best result was finally achieved at high temperature in DMF, using 3 equiv. of DBU. The expected product (**3**) was formed in the reaction with a yield of 33%.

Since we also wanted to investigate the biological effect of the free amidinium salts, we attempted their direct production based on literature examples [23,35], by rapid synthesis without the use of protecting groups (Figure 3). For this rapid synthetic method, we first synthesized the 6-*O*-tosyl derivative of methyl α-d-glucopyranoside (**40**) [36,37], then reacted this compound with DBU in dry acetonitrile. Neither the expected DBU-conjugated derivative nor the 5,6-eliminated derivative was formed in the reaction. The 3,6-anhydro derivative of the methyl α-d-glucopyranoside (**41**) [38] was identified as the main product, which was formed in excellent yield (80%). We also attempted to prepare the unprotected derivative from the 6-OTs derivative of phenyl 1-thio-α-d-mannopyranoside (**42**) [39,40]. We tested two types of reaction conditions (a, CH_3_CN + DBU and b, DMF + DBU), but unfortunately neither of the reactions produced the expected product. However, similar to the previous reaction, the 3,6-anhydro derivative of phenylthio-mannoside (**43**) was formed with excellent yield (97%). Finally, we attempted to carry out the reaction from the 6-iodo derivative of methyl α-d-glucopyranoside (**44**) [41], but these reactions did not lead to results either: here again, the 3,6-anhydro derivative (**41**) was formed along with other unidentified decomposition products. Based on these failed experiments, it can be concluded that the planned DBU-conjugated derivatives cannot be produced directly from the free derivatives under the conditions tested. Protection of the -OH groups of the carbohydrate part is essential for successful nucleophilic substitution reactions.

Based on the preliminary biological tests, we removed certain protective groups from the protected derivatives that seemed promising, thus examining the effect of each group on the biological activity, and in this way we were also able to increase the water solubility of our compounds. In initial studies, d-*gluco* and d-*manno* derivatives appeared to be active. Based on this, we first transformed the compounds with the d-*gluco* configuration (Figure 4). The benzoyl groups were removed from compounds **3** and **11** under Zemplén conditions, thus releasing the hydroxyls at position 2 and 3 (**45**, **46**). The 2-naphthylmethyl group (2-NAP) was cleaved from derivatives **10** and **46** in CH_2_Cl_2_/H_2_O using 2,3-dichloro-5,6-dicyano-1,4-benzoquinone (DDQ) reagent, thus reaching compounds **47** and **48**.

The d-*manno* derivatives were further modified in a similar manner (Figure 5). The 2-NAP group was removed from derivatives **13**, **14** and **16** under oxidative conditions using DDQ, thus releasing one -OH per molecule (**49**, **50**, **52**). Furthermore, the benzoyl groups were removed from compounds **15** and **16**, thus reaching derivatives **51** and **53**.

After that, the antifungal and antibacterial effects of the synthesized derivatives were investigated, and the cytotoxicity, hemolytic activity and membrane permeabilizing effect of some compounds were also determined.

### 2.2. Biological Investigations of the DBU-Conjugated Derivatives

#### 2.2.1. Antifungal and Antibacterial Studies

The antimicrobial activity of twenty-two cationic hexopyranoside-amidinium salts (**3**, **10**, **11**, **12**, **21**, **22**, **23**, **24**, **29**, **30**, **32**, **37**, **38**, **45**, **46**, **47**, **48**, **49**, **50**, **51**, **52** and **53**) were compared in susceptibility tests (Table 2). The effectiveness of all compounds was tested on *Staphylococcus aureus* ssp. *aureus* (ATCC 6538), *Escherichia coli* (ATCC 8739) and *Candida albicans* (ATCC 10231). We tested the antimicrobial activity of the compounds at the concentration range of 6.25–100 µg/mL.

Compounds **10**, **12** and **22** showed a strong growth inhibitory effect on Gram-positive bacterium, *Staphylococcus aureus* ssp. *aureus*, their MIC values being smaller than 6.25 µg/mL (Table 2). Compounds **3**, **21**, **24**, **29**, **30**, **32**, **37**, **48**, 50, **51**, and **53** also showed activity against the *S aureus* strain with MIC values of 6.25–50 µg/mL. At the same time, compounds **11**, **23**, **38**, **45**, **46**, **47**, **49** and **52** had no anti Gram-positive effect even at 100 µg/mL concentration.

Compounds **10**, **22** and **29** had an inhibitory effect on yeast, *Candida albicans*, with MIC values lower than 6.25 µg/mL (Table 2). MIC values of **3**, **12**, **21**, **24**, **30**, **48**, **50**, **51** and **53** were in the range 6.25–50µg/mL on *Candida albicans,* while **11**, **23**, **32**, **37**, **38**, **45**, **46**, **47**, **49** and **52** were effective in concentration higher than 100 µg/mL.

The Gram-negative bacteria, *Escherichia coli,* was less sensitive to the cationic carbohydrate derivatives. None of the tested compounds had any effect on *Escherichia coli* in the highest used concentration, 100 µg/mL.

Compounds **10** and **22** were the most effective derivatives. Compounds **11**, **23**, **38**, **45**, **46**, **47**, **49** and **52** had no effect on any tested microorganism in concentrations as high as 100 µg/mL.

The activity of the three most effective antifungal compounds was also determined at lower concentrations (10× diluted) (Table 3). Testing the activity of the compounds against *C. albicans* at the concentration range of 0.625–10.0 µg/mL, we found that compounds **22** and **29** have excellent antifungal activity with 2.5 µg/mL MIC values, and derivative **10** is effective at a concentration of 5 µg/mL.

#### 2.2.2. Cytotoxic Activity

The cytotoxic effect of six selected compounds (**3**, **10**, **12**, **21, 22** and **29**) with promising antimicrobial activity was studied on *HaCaT* (human keratinocyte) cell lines. Using the MTT-test, we determined the viability of the cells (Figure 3), and the half maximal inhibitory concentration (IC_50_) values of the compounds are summarized in Table 4. The duration of tests was 24 h; the examined concentrations were: 50–25–12.5–6.25–3.125 µg/mL.

Among the tested DBU-sugar conjugates, compound **3** showed the strongest cytotoxic effect on *HaCaT* cell line (Figure 3A), and its IC_50_ value was as low as 0.13 µg/mL. Compounds **12** and **21** also had significant toxic effect (Figure 3B,C) with IC_50_ values of 4.26 and 1.02 µg/mL, respectively.

Compounds **10**, **22**, **29** possessed only slight cytotoxicity (Figure 3D–F). Their IC_50_ values were one order of magnitude higher than that of compounds **12** and **21** and two orders of magnitude higher than that of compound **3**. Most importantly, compounds **10** and **22** hardly affected the cell viability at their antibacterially and antifungally active concentrations of ≤6.25 µg/mL (Figure 3D,E).

#### 2.2.3. Hemolytic Activity

The hemolytic activity of compounds **10** and **22** was determined on freshly isolated rat red blood cells (Figure 4). Both compounds showed a dose-dependent hemolytic activity. At 6.25 µg/mL they did not cause significant hemolysis compared to the negative control PBS (*p* > 0.05); however, at 50 and 100 µg/mL they caused significant haemoglobin release (*p* < 0.0001) and their effect were comparable to the hemolytic effect of water used as positive control.

#### 2.2.4. Membrane Permeabilization Study

In membrane permeabilization experiments, the Gram-positive *Lactobacillus plantarum* was used to test the effect of compounds **10** and **22** at 6.25 µg/mL for 48 h. Bacterial cells were stained with SYTOX green (SG) fluorescent dye, which is impermeable on intact membranes but easily penetrates through compromised membranes. Test compounds **10** and **22** increased the ratio of SG-positive cells after 1 h of incubation and drastically increased it after 4 h of incubation (Figure 5), showing their membrane-compromising and toxic effects on *Lactobacillus plantarum.*

## 3. Materials and Methods

### 3.1. General Information about the Syntheses

Optical rotations were measured at room temperature on a Perkin-Elmer 241 automatic polarimeter (PerkinElmer, Waltham, MA, US). TLC analysis was performed on Kieselgel 60 F_254_ (Merck) silica-gel plates with visualization by immersing in a sulfuric-acid solution (5% in EtOH) followed by heating. Column chromatography was performed on silica gel 60 (Merck 0.063–0.200 mm). Organic solutions were dried over MgSO_4_ and concentrated under vacuum. ^1^H and J-modulated ^13^C NMR spectroscopy (^1^H: 400 and 500 MHz; ^13^C: 100.28 and 125.76 MHz) were performed on Bruker DRX-400 and Bruker Avance II 500 spectrometers at 25 °C (Bruker, Billerica, MA, USA) Chemical shifts are referenced to SiMe_4_ or sodium 3-(trimethylsilyl)-1-propanesulfonate (DSS, *δ* = 0.00 ppm for ^1^H nuclei) and to residual solvent signals (CDCl_3_: *δ* = 77.16 ppm, CD_3_OD: *δ* = 49.15 ppm for ^13^C nuclei). You can find the NMR spectra of the new compounds in the Appendix A. HRMS measurements were carried out on a maXis II UHR ESI-QTOF MS instrument (Bruker, Billerica, MA, USA) in positive ionization mode. The following parameters were applied for the electrospray ion source: capillary voltage: 3.6 kV; end plate offset: 500 V; nebulizer pressure: 0.5 bar; dry gas temperature: 200 °C; and dry gas flow rate: 4.0 L/min. Constant background correction was applied for each spectrum. The background was recorded before each sample by injecting the blank sample matrix (solvent). Na-formate calibrant was injected after each sample, which enabled internal calibration during data evaluation. Mass spectra were recorded by otofControl version 4.1 (build: 3.5, Bruker) and processed by Compass DataAnalysis version 4.4 (build: 200.55.2969).

#### 3.1.1. General Methods

General **Method A** for Zemplén debenzoylation (**45**, **46**, **51** and **53**)

To a stirred solution of appropriate benzoyl protected compounds (**3**, **11**, **15** and **16**) (0.377 mmol) in MeOH (7.5 mL), NaOMe (25 mg, 0.472 mmol, 1.25 equiv.) was added and the reaction mixture was stirred at room temperature for 24 h. After 24 h, the mixture was neutralized with AcOH (500 µL) and the mixture was concentrated under reduced pressure.

2.General **Method B** for cleavage of the 2-naphthylmethyl-group (NAP) (**47–50** and **52**)

The corresponding NAP-ether protected compounds (**10**, **13**, **14**, **16**, and **46**) (0.125 mmol) was dissolved in the mixture of CH_2_Cl_2_ (2.7 mL) and water (300 μL) then DDQ was added (71 mg, 0.313 mmol, 2.5 equiv.) and the reaction mixture was stirred at room temperature for 30 min. After 30 min, the mixture was neutralized with Cl¯ exchange resin, filtered, washed with CH_2_Cl_2_ and concentrated under reduced pressure.

#### 3.1.2. Optimization of the Synthesis of the DBU-Conjugated Methyl α-d-glucopyranoside derivative 3

Methyl 2,3-di-*O*-benzoyl-4-*O*-benzyl-α-d-*xylo*-hex-5-enopyranoside (**2**) [28], 8-*N*-[methyl 6-deoxy-6-yl-2,3-di-*O*-benzoyl-4-*O*-benzyl-α-d-glucopyranoside]-1,8-diazabicyclo(5.4.0)undec-7-ene-iodide (**3**) [28] and methyl 6-*O*-acetyl-2,3-di-*O*-benzoyl-4-*O*-benzyl-α-d-glucopyranoside (**39**).

**Reaction 1**: To the solution of compound **1 [28]**, (203 mg, 0.337 mmol) in dry CH_3_CN (2.0 mL), DBU (50 μL, 0.337 mmol, 1.0 equiv.) was added. The reaction mixture was stirred at room temperature for 16 days. After 16 days, the reaction mixture was evaporated under reduced pressure. The crude product was purified by silica gel chromatography (9:1 CH_2_Cl_2_/*n*-hexane, then CH_2_Cl_2_ and finally 9:1 CH_2_Cl_2_/MeOH) to give **1** (18 mg, 9%) as a colorless syrup, **2** (101 mg, 63%) as a white crystal, and **3** (27 mg, 11%) as yellow foam.

**Reaction 2**: To the solution of compound **1** (503 mg, 0.835 mmol) in dry CH_3_CN (5.0 mL), DBU (187 μL, 1.253 mmol, 1.5 equiv.) was added. The reaction mixture was stirred at room temperature for 4 days. After 4 days, the reaction mixture was evaporated under reduced pressure. The crude product was purified by silica gel chromatography (CH_2_Cl_2_, then 9:1 CH_2_Cl_2_/MeOH) to give **2** (284 mg, 72%) as a colorless syrup and **3** (90 mg, 14%) as a yellow foam.

**Reaction 3**: To the solution of compound **1** (290 mg, 0.482 mmol) in dry CH_3_CN (2.9 mL), DBU (216 μL, 1.446 mmol, 3.0 equiv.) and glacial acetic acid (82 μL, 1.446 mmol, 3.0 equiv.) were added. The reaction mixture was stirred at room temperature. After 3 days, the reaction mixture was evaporated under reduced pressure. The crude product was purified by silica gel chromatography (CH_2_Cl_2_, then 99:1 CH_2_Cl_2_/acetone, then 9:1 CH_2_Cl_2_/MeOH) to give **2** (23 mg, 10%) as a colorless syrup and **39** (223 mg, 87%) as a white foam.

Data of **39**: [α]_D_^25^+150.7 (*c* 0.15, CDCl_3_); *R*_f_ 0.43 (CH_2_Cl_2_); ^1^H NMR (500 MHz, CDCl_3_) *δ* = 8.01–7.15 (m, 15H, arom.), 6.07 (t, *J* = 9.4 Hz, 1H, H-3), 5.17 (dd, *J* = 3.5 Hz, *J* = 10.2 Hz, 1H, H-2), 5.12 (d, *J* = 3.5 Hz, 1H, H-1), 4.63 (d, *J* = 10.9 Hz, 1H, BnC*H*_2_a), 4.54 (d, *J* = 10.9 Hz, 1H, BnC*H*_2_b), 4.41 (t, *J* = 12.0 Hz, 1H, H-6a), 4.34 (dd, *J* = 4.0 Hz, *J* = 12.1 Hz, 1H, H-6b), 4.07–4.04 (m, 1H, H-5), 3.86 (t, *J* = 9.4 Hz, 1H, H-4), 3.40 (s, 3H, OC*H*_3_), 2.11 (s, 3H, Ac-C*H*_3_) ppm; ^13^C NMR (125 MHz, CDCl_3_) *δ* = 170.7 (1C, Ac C=O), 166.1, 165.6 (2C, 2 × Bz C=O), 137.1, 129.7, 129.1 (3C, 3 × C_q_ arom.), 133.4–128.1 (15C, arom.), 97.0 (1C, C-1), 75.8 (1C, C-4), 74.7 (1C, Bn*C*H_2_), 72.7 (1C, C-3), 72.2 (1C, C-2), 68.6 (1C, C-5), 62.8 (1C, C-6), 55.5 (1C, O*C*H_3_), 21.0 (1C, Ac-*C*H_3_) ppm; UHR ESI-QTOF: m/z calcd for C_30_H_30_NaO_9_ [M+Na]^+^ 557.1782; found: 557.1777.

**Reaction 4**: To the solution of compound **1** (305 mg, 0.506 mmol) in dry CH_3_CN (3.0 mL), DBU (454 μL, 3.034 mmol, 6.0 equiv.) and *p*-TSA (577 mg, 3.034 mmol, 6.0 equiv.) were added. The reaction mixture was stirred at room temperature for 21 h, then for 5 h at 85 °C. After 26 h, the reaction mixture was evaporated under reduced pressure. The crude product was purified by silica gel chromatography (CH_2_Cl_2_) to give **1** (256 mg, 84%) as a colorless syrup. The starting material was recovered.

**Reaction 5**: To the solution of compound **1** (528 mg, 0.877 mmol) in dry CH_3_CN (5.3 mL), DBU (787 μL, 5.261 mmol, 6.0 equiv.) was added. The reaction mixture was stirred at room temperature for 6 days. After 6 days, the reaction mixture was evaporated under reduced pressure. The crude product was purified by silica gel chromatography (CH_2_Cl_2_, then 9:1 CH_2_Cl_2_/MeOH) to give **2** (292 mg, 70%) as a white crystal and **3** (120 mg, 18%) as a yellow foam.

**Reaction 6**: To the compound **1** (254 mg, 0.422 mmol), DBU (2.0 mL, 13.2 mmol, 31.25 equiv.) was added at 0 °C. The reaction mixture was stirred at room temperature for 24 h. After 24 h, the reaction mixture was evaporated under reduced pressure. The crude product was purified by silica gel chromatography (CH_2_Cl_2_, then 9:1 CH_2_Cl_2_/MeOH) to give **2** (64 mg, 32%) as a colorless syrup, **3** (36 mg, 11%) as a yellow foam, and other, not isolated, degradation products.

**Reaction 7**: To the solution of compound **1** (288 mg, 0.479 mmol) in dry DMF (2.9 mL) heated to 150 °C, DBU (215 μL, 1.437 mmol, 3.0 equiv.) was added. The reaction mixture was stirred at 150 °C for 45 min. After 45 min, the reaction mixture was evaporated under reduced pressure. The crude product was purified by silica gel chromatography (CH_2_Cl_2_, then 9:1 CH_2_Cl_2_/MeOH) to give **1** (5 mg, 2%) as a colorless syrup, **2** (133 mg, 58%) as a white crystal, and **3** (119 mg, 33%) as a yellow foam.

Methyl 3,6-anhydro-α-d-glucopyranoside (**41**) [38].

**Reaction 1**: To the solution of compound **40 [36,37]** (500 mg, 1.436 mmol) in dry CH_3_CN (8.5 mL), DBU (1288 μL, 8.618 mmol, 6.0 equiv.) was added. The reaction mixture was stirred at room temperature for 20 h and at 85 °C for 4 h. After 24 h the reaction mixture was evaporated under reduced pressure. The crude product was purified by silica gel chromatography (97:3 to 95:5 CH_2_Cl_2_/MeOH, then MeOH + 1% 60% aqueous solution of AcOH) to give **41 [38]** (202 mg, 80%) as a colorless syrup and other, not isolated, degradation products.

**Reaction 2**: To the solution of compound **44 [41]** (413 mg, 1.359 mmol) in dry CH_3_CN (8.1 mL), DBU (1219 μL, 8.152 mmol, 6.0 equiv.) was added. The reaction mixture was stirred at room temperature for 2 days at room temperature. After 2 days, the reaction mixture was evaporated under reduced pressure. The crude product was purified by silica gel chromatography (9:1 CH_2_Cl_2_/MeOH, then MeOH + 1% 60% aqueous solution of AcOH) to give **41 [38]** (151 mg,62%) as a colorless syrup and other, not isolated, degradation products.

**Reaction 3**: To the solution of compound **44** (406 mg, 1.336 mmol) in dry DMF (8.1 mL) heated to 145 °C, DBU (599 μL, 4.008 mmol, 3.0 equiv.) was added. The reaction mixture was stirred at 145 °C for 30 min. After 30 min, the reaction mixture was evaporated under reduced pressure. The crude product was purified by silica gel chromatography (9:1 CH_2_Cl_2_/MeOH, then MeOH + 1% 60% aqueous solution of AcOH) to give **41** (136 mg, 58%) as a colorless syrup and other, not isolated, degradation products.

Phenyl 3,6-anhydro-1-thio-α-d-mannopyranoside (**43**)

**Reaction 1**: To the solution of compound **42 [39,40]** (483 mg, 1.134 mmol) in dry CH_3_CN (6.8 mL), DBU (1018 μL, 6.804 mmol, 6.0 equiv.) was added. The reaction mixture was stirred at room temperature for 5 days. After 5 days, the reaction mixture was evaporated under reduced pressure. The crude product was purified by silica gel chromatography (9:1 CH_2_Cl_2_/MeOH, then MeOH + 1% 60% aqueous solution of AcOH) to give **43** (279 mg, 97%) as a white foam. [α]_D_^25^+85.5 (*c* 0.20, MeOH); *R*_f_ 0.56 (9:1 CH_2_Cl_2_/MeOH); ^1^H NMR (400 MHz, CDCl_3_) *δ* = 7.59–7.27 (m, 5H, arom.), 4.94 (d, *J* = 8.8 Hz, 1H, H-1), 4.28–4.27 (m, 1H, H-5), 4.26–4.25 (m, 1H, H-3), 4.21 (s, 1H, H-4), 4.11 (d, *J* = 11.0 Hz, 1H, H-6a), 3.98 (dd, *J* = 2.5 Hz, *J* = 10.9 Hz, 1H, H-6b), 3.78 (d, *J* = 8.6 Hz, 1H, H-2), 3.12–3.09 (m, 1H, H-2-O*H*), 2.86–2.83 (m, 1H, H-3-O*H*) ppm; ^13^C NMR (100 MHz, CDCl_3_) *δ* = 132.5 (1C, 1 × C_q_ arom.), 132.8–128.3 (5C, arom.), 85.3 (1C, C-1), 77.8 (1C, C-3), 76.8 (1C, C-5), 71.3 (1C, C-4), 68.9 (1C, C-6), 67.6 (1C, C-2) ppm; UHR ESI-QTOF: m/z calcd for C_12_H_14_NaO_4_S [M+Na]^+^ 277.0505; found: 277.0494.

**Reaction 2**: To the solution of compound **42** (497 mg, 1.167 mmol) in dry DMF (7.1 mL) heated to 145 °C, DBU (524 μL, 3.501 mmol, 3.0 equiv.) was added. The reaction mixture was stirred at 145 °C for 4 h. After 4 h, the reaction mixture was evaporated under reduced pressure. The crude product was purified by silica gel chromatography (9:1 CH_2_Cl_2_/MeOH, then MeOH + 1% 60% aqueous solution of AcOH) to give **43** (275 mg, 93%) as a white foam.

#### 3.1.3. Synthesis and Transformation of the DBU-Conjugated Derivatives

*8-N-[methyl 6-deoxy-6-yl-4-O-benzyl-α-d-glucopyranoside]-1,8-diazabicyclo(5.4.0)undec-7-ene-chloride (**45**).* Compound **3 [28]** (110 mg, 0.146 mmol) was converted to **45** according to general **Method A**. The crude product was purified by silica gel chromatography (75:25 CH_2_Cl_2_/MeOH) to give **45** (48 mg, 73%) as a colorless syrup. [α]_D_^25^+60.0 (*c* 0.19, MeOH); *R*_f_ 0.63 (75:25 CH_2_Cl_2_/MeOH); ^1^H NMR (500 MHz, CD_3_OD) *δ* = 7.41–7.28 (m, 5H, arom.), 5.05 (d, *J* = 11.1 Hz, 1H, BnC*H*_2_a), 4.70 (d, *J* = 3.7 Hz, 1H, H-1), 4.65 (d, *J* = 11.2 Hz, 1H, BnC*H*_2_b), 3.85–3.82 (m, 2H, H-3, H-5), 3.75–3.51 (m, 8H, H-6a,b, 3 × NC*H*_2_ DBU), 3.46 (dd, *J* = 3.7 Hz, *J* = 9.8 Hz, 1H, H-2), 3.38 (s, 3H, OC*H*_3_), 3.27 (t, *J* = 8.8 Hz, 1H, H-4), 2.88 (t, *J* = 5.5 Hz, 2H, C*H*_2_ DBU), 2.07 (dq, *J* = 6.4 Hz, *J* = 13.1 Hz, 2H, C*H*_2_ DBU), 1.77–1.58 (m, 6H, 3 × C*H*_2_ DBU) ppm; ^13^C NMR (125 MHz, CD_3_OD) *δ* = 168.8 (1C, C_q_ DBU), 139.7 (1C, C_q_ arom.), 129.6–128.9 (5C, arom.), 101.3 (1C, C-1), 80.8 (1C, C-4), 75.8 (1C, Bn*C*H_2_), 75.4 (1C, C-5), 73.3 (1C, C-2), 71.1 (1C, C-3), 56.4 (1C, O*C*H_3_), 56.2 (1C, N*C*H_2_ DBU), 56.0 (1C, C-6), 50.3, 49.4 (2C, 2 × N*C*H_2_ DBU), 29.4, 29.3, 27.0, 24.1, 21.3 (5C, 5 × *C*H_2_ DBU) ppm; UHR ESI-QTOF: m/z calcd for C_23_H_35_N_2_O_5_ [M]^+^ 419.2540; found: 419.2537.

*8-N-[ethyl 6-deoxy-6-yl-4-O-(2′-naphthyl)methyl-1-thio-α-d-glucopyranoside]-1,8-diazabicyclo(5.4.0)undec-7-ene-chloride (**46**).* Compound **11 [28]** (315 mg, 0.377 mmol) was converted to **46** according to general **Method A**. The crude product was purified by silica gel chromatography (9:1 CH_2_Cl_2_/MeOH) to give **46** (190 mg, 91%) as a light yellow syrup. [α]_D_^25^+120.8 (*c* 0.13, MeOH); *R*_f_ 0.40 (9:1 CH_2_Cl_2_/MeOH); ^1^H NMR (500 MHz, CD_3_OD) *δ* = 7.88–7.46 (m, 7H, arom.), 5.35 (d, *J* = 5.4 Hz, 1H, H-1), 5.20 (d, *J* = 11.2 Hz, 1H, NAPC*H*_2_a), 4.84 (d, *J* = 11.3 Hz, 1H, NAPC*H*_2_b), 4.13 (t, *J* = 9.4 Hz, 1H, H-5), 3.86–3.81 (m, 2H, H-2, H-6a), 3.74 (t, *J* = 9.1 Hz, 1H, H-3), 3.67 (dd, *J* = 9.6 Hz, *J* = 15.4 Hz, 1H, H-6b), 3.57–3.38 (m, 6H, 3 × NC*H*_2_ DBU), 3.35 (t, *J* = 9.1 Hz, 1H, H-4), 2.78–2.74 (m, 2H, C*H*_2_ DBU), 2.52 (dd, *J* = 7.2 Hz, *J* = 14.5 Hz, 2H, SEt, C*H*_2_), 1.98–1.95 (m, 2H, C*H*_2_ DBU), 1.63–1.49 (m, 6H, 3 × C*H*_2_ DBU), 1.25 (t, *J* = 7.4 Hz, 3H, SEt, C*H*_3_) ppm; ^13^C NMR (125 MHz, CD_3_OD) *δ* = 168.6 (1C, C_q_ DBU), 137.1, 134.6, 134.3 (3C, 3 × C_q_ arom.), 129.3–127.1 (7C, arom.), 87.0 (1C, C-1), 80.8 (1C, C-4), 76.1 (1C, C-3), 75.7 (1C, NAP*C*H_2_), 72.7 (1C, C-2), 71.3 (1C, C-5), 56.1 (1C, N*C*H_2_ DBU), 56.0 (1C, C-6), 50.2, 49.3 (2C, 2 × N*C*H_2_ DBU), 29.3, 29.1, 26.9 (3C, 3 × *C*H_2_ DBU), 25.9 (1C, SEt *C*H_2_), 24.0, 21.0 (2C, 2 × *C*H_2_ DBU), 15.5 (1C, SEt *C*H_3_) ppm; UHR ESI-QTOF: m/z calcd for C_28_H_39_N_2_O_4_S [M]^+^ 499.2625; found: 499.2624.

*8-N-[ethyl 6-deoxy-6-yl-1-thio-α-d-glucopyranoside]-1,8-diazabicyclo(5.4.0)undec-7-ene-chloride (**47**).* Compound **46** (70 mg, 0.125 mmol) was converted to **47** according to general **Method B**. The crude product was purified by silica gel chromatography (7:3 CH_2_Cl_2_/MeOH) to give **47** (38 mg, 78%) as a light yellow syrup. [α]_D_^25^+142.7 (*c* 0.15, MeOH); *R*_f_ 0.37 (7:3 CH_2_Cl_2_/MeOH); ^1^H NMR (500 MHz, CD_3_OD) *δ* = 5.35 (d, *J* = 5.2 Hz, 1H, H-1), 4.13 (t, *J* = 8.8 Hz, 1H, H-5), 4.00 (d, *J* = 15.5 Hz, 1H, H-6a), 3.80 (dd, *J* = 9.1 Hz, *J* = 15.6 Hz, 1H, H-6b), 3.72–3.58 (m, 7H, H-2, 3 × NC*H*_2_ DBU), 3.46 (t, *J* = 9.2 Hz, 1H, H-3), 3.19 (t, *J* = 9.1 Hz, 1H, H-4), 3.03 (dd, *J* = 7.7 Hz, *J* = 15.5 Hz, 1H, C*H*_2_a DBU), 2.95 (dd, *J* = 9.5 Hz, *J* = 15.2 Hz, 1H, C*H*_2_b DBU), 2.57 (td, *J* = 7.2 Hz, *J* = 11.5 Hz, 2H, SEt, C*H*_2_), 2.09 (dd, *J* = 6.0 Hz, *J* = 11.6 Hz, 2H, C*H*_2_ DBU), 1.81–1.73 (m, 6H, 3 × C*H*_2_ DBU), 1.29 (t, *J* = 7.4 Hz, 3H, SEt, C*H*_3_) ppm; ^13^C NMR (125 MHz, CD_3_OD) *δ* = 169.1 (1C, C_q_ DBU), 87.5 (1C, C-1), 75.5 (1C, C-3), 73.3 (1C, C-4), 72.7 (1C, C-2), 72.1 (1C, C-5), 56.1 (1C, N*C*H_2_ DBU), 55.9 (1C, C-6), 50.3, 49.5 (2C, 2 × N*C*H_2_ DBU), 29.4, 27.1 (3C, 3 × *C*H_2_ DBU), 25.7 (1C, SEt *C*H_2_), 24.3, 21.2 (2C, 2 × *C*H_2_ DBU), 15.4 (1C, SEt *C*H_3_) ppm; UHR ESI-QTOF: m/z calcd for C_17_H_31_N_2_O_4_S [M]^+^ 359.1999; found: 359.1999.

*8-N-[ethyl 2,3-di-O-benzyl-6-deoxy-6-yl-1-thio-α-d-glucopyranoside]-1,8-diazabicyclo(5.4.0)undec-7-ene-chloride (**48**).* Compound **10 [28]** (218 mg, 0.270 mmol) was converted to **48** according to general **Method B**. The crude product was purified by silica gel chromatography (85:15 CH_2_Cl_2_/MeOH) to give **48** (98 mg, 67%) as a light yellow syrup. [α]_D_^25^+88.3 (*c* 0.12, CHCl_3_); *R*_f_ 0.42 (85:15 CH_2_Cl_2_/MeOH); ^1^H NMR (500 MHz, CDCl_3_) *δ* = 7.40–7.21 (m, 10H, arom.), 5.74–7.71 (m, 1H, H-4-O*H*), 5.22 (d, *J* = 2.5 Hz, 1H, H-1), 4.95 (d, *J* = 11.3 Hz, 1H, BnC*H*_2_a), 4.83, (d, *J* = 11.3 Hz, 1H, BnC*H*_2_b), 4.68–4.63 (m, 2H, BnC*H2*), 4.43 (d, *J* = 15.3 Hz, 1H, H-6a), 4.16 (t, *J* = 8.7 Hz, 1H, H-5), 3.69–3.68 (m, 2H, H-2, H-3), 3.58–3.54 (m, 7H, H-6a, 3 × NC*H*_2_ DBU), 3.47–3.45 (m, 1H, H-4), 3.01–2.96 (m, 1H, C*H*_2_a DBU), 2.92–2.88 (m, 1H, C*H*_2_b DBU), 2.49–2.41 (m, 2H, SEt, C*H*_2_), 2.01 (s, 2H, C*H*_2_ DBU), 1.72–1.60 (m, 6H, 3 × C*H*_2_ DBU), 1.26 (t, *J* = 7.3 Hz, 3H, SEt, C*H*_3_) ppm; ^13^C NMR (125 MHz, CDCl_3_) *δ* = 167.6 (1C, C_q_ DBU), 139.3, 138.0 (2C, 2 × C_q_ arom.), 128.4–127.4 (10C, arom.), 83.8 (1C, C-1), 81.7 (1C, C-3), 78.6 (1C, C-2), 75.3, 72.6 (2C, 2 × Bn*C*H_2_), 72.0 (1C, C-5), 71.8 (1C, C-4), 55.5 (1C, N*C*H_2_ DBU), 55.4 (1C, C-6), 49.4, 48.5 (2C, 2 × N*C*H_2_ DBU), 28.8, 28.4, 26.2 (3C, 3 × *C*H_2_ DBU), 24.4 (1C, SEt *C*H_2_), 23.3, 20.2 (2C, 2 × *C*H_2_ DBU), 14.9 (1C, SEt *C*H_3_) ppm; UHR ESI-QTOF: m/z calcd for C_31_H_43_N_2_O_4_S [M]^+^ 539.2938; found: 539.2939.

*8-N-[phenyl 2,4-di-O-benzyl-6-deoxy-6-yl-1-thio-α-d-mannopyranoside]-1,8-diazabicyclo(5.4.0)undec-7-ene-chloride (**49**).* Compound **13 [27]** (119 mg, 0.140 mmol) was converted to **49** according to general **Method B**. The crude product was purified by silica gel chromatography (8:2 CH_2_Cl_2_/MeOH) to give **49** (30 mg, 60%) as a light yellow syrup. [α]_D_^25^+84.5 (*c* 0.11, CHCl_3_); *R*_f_ 0.50 (8:2 CH_2_Cl_2_/MeOH); ^1^H NMR (500 MHz, CDCl_3_) *δ* = 7.39–7.25 (m, 15H, arom.), 5.63 (s, 1H, H-1), 4.98–4.64 (m, 4H, 2 × BnC*H*_2_), 4.02–3.99 (m, 2H, H-3, H-5), 3.96 (s, 1H, H-2), 3.85–3.82 (m, 1H, H-6a), 3.61 (s, 1H, H-3-O*H*), 3.51–3.49 (m, 2H, H-4, H-6b), 3.35–3.31 (m, 6H, 3 × NC*H*_2_ DBU), 2.63 (s, 2H, C*H*_2_ DBU), 1.72–1.22 (m, 8H, 4 × C*H*_2_ DBU) ppm; ^13^C NMR (125 MHz, CDCl_3_) *δ* = 167.3 (1C, C_q_ DBU), 137.9, 137.4, 133.1 (3C, 3 × C_q_ arom.), 130.7–127.8 (15C, arom.), 83.9 (1C, C-1), 79.3 (1C, C-2), 77.8 (1C, C-4), 75.4, 73.2 (2C, 2 × Bn*C*H_2_), 72.2 (1C, C-5), 71.3 (1C, C-3), 55.9 (1C, N*C*H_2_ DBU), 55.2 (1C, C-6), 49.8, 49.7 (2C, 2 × N*C*H_2_ DBU), 29.8, 28.9, 26.1, 25.2, 23.1 (5C, 5 × *C*H_2_ DBU) ppm; UHR ESI-QTOF: m/z calcd for C_35_H_43_N_2_O_4_S [M]^+^ 587.2938; found: 587.2938.

*8-N-[phenyl 2,3-di-O-benzyl-6-deoxy-6-yl-1-thio-α-d-mannopyranoside]-1,8-diazabicyclo(5.4.0)undec-7-ene-chloride (**50**).* Compound **14 [27]** (59 mg, 0.069 mmol) was converted to **50** according to general **Method B**. The crude product was purified by silica gel chromatography (8:2 CH_2_Cl_2_/MeOH) to give **50** (32 mg, 76%) as a light yellow syrup. [α]_D_^25^+73.0 (*c* 0.10, CHCl_3_); *R*_f_ 0.45 (8:2 CH_2_Cl_2_/MeOH); ^1^H NMR (500 MHz, CDCl_3_) *δ* = 7.41–7.24 (m, 15H, arom.), 5.58 (s, 1H, H-1), 4.92–4.61 (m, 4H, 2 × BnC*H*_2_), 4.40 (d, *J* = 15.4 Hz, 1H, H-6a), 4.24 (t, *J* = 8.5 Hz, 1H, H-5), 3.96–3.92 (m, 2H, H-2, H-4), 3.90–3.88 (m, 1H, H-3), 3.72–3.34 (m, 7H, H-6b, 3 × NC*H*_2_ DBU), 2.94–2.89 (m, 1H, C*H*_2_a DBU), 2.76–2.71 (m, 1H, C*H*_2_b DBU), 1.81–1.53 (m, 8H, 4 × C*H*_2_ DBU), 1.23 (t, *J* = 7.0 Hz, 1H, H-4-O*H*) ppm; ^13^C NMR (125 MHz, CDCl_3_) *δ* = 167.6 (1C, C_q_ DBU), 138.9, 138.2, 134.1 (3C, 3 × C_q_ arom.), 130.7–127.2 (15C, arom.), 85.3 (1C, C-1), 79.3 (1C, C-3), 77.3 (1C, C-2), 73.6 (1C, C-5), 73.4, 73.1 (2C, 2 × Bn*C*H_2_), 68.9 (1C, C-4), 55.5 (1C, N*C*H_2_ DBU), 55.4 (1C, C-6), 49.4, 48.3 (2C, 2 × N*C*H_2_ DBU), 28.8, 28.5, 26.0, 23.2, 20.0 (5C, 5 × *C*H_2_ DBU) ppm; UHR ESI-QTOF: m/z calcd for C_35_H_43_N_2_O_4_S [M]^+^ 587.2938; found: 587.2937.

*8-N-[phenyl 6-deoxy-6-yl-4-O-(2′-naphthyl)methyl-1-thio-α-d-mannopyranoside]-1,8-diazabicyclo(5.4.0)undec-7-ene-chloride (**51**).* Compound **15 [27]** (51 mg, 0.057 mmol) was converted to **51** according to general **Method A**. The crude product was purified by silica gel chromatography (85:15 CH_2_Cl_2_/MeOH) to give **51** (28 mg, 84%) as a light yellow syrup. [α]_D_^25^+153.0 (*c* 0.10, MeOH); *R*_f_ 0.37 (9:1 CH_2_Cl_2_/MeOH); ^1^H NMR (500 MHz, CDCl_3_) *δ* = 7.96–7.19 (m, 12H, arom.), 5.88 (s, 1H, H-1), 5.23 (d, *J* = 11.2 Hz, 1H, NAPC*H*_2_a), 4.82 (d, *J* = 11.3 Hz, 1H, NAPC*H*_2_b), 4.33 (s, 1H, H-2), 4.13–4.08 (m, 2H, H-3, H-5), 4.00–3.94 (m, 2H, H-4, H-6a), 3.69 (d, *J* = 14.2 Hz, 1H, H-6b), 3.46–3.22 (m, 6H, 3 × NC*H*_2_ DBU), 2.78–2.75 (m, 1H, C*H*_2_a DBU), 2.66–2.63 (m, 1H, C*H*_2_b DBU), 2.45 (d, *J* = 3.8 Hz, 1H, H-3-O*H*), 1.64–1.26 (m, 9H, H-2-O*H*, 4 × C*H*_2_ DBU) ppm; ^13^C NMR (125 MHz, CD_3_OD) *δ* = 168.5 (1C, C_q_ DBU), 137.1, 134.8, 134.5, 134.0 (4C, 4 × C_q_ arom.), 132.1–127.2 (12C, arom.), 88.0 (1C, C-1), 78.2 (1C, C-4), 76.1 (1C, NAP*C*H_2_), 73.6 (1C, C-3), 73.3 (1C, C-2), 72.3 (1C, C-5), 56.0 (1C, N*C*H_2_ DBU), 55.8 (1C, C-6), 50.0, 49.3 (2C, 2 × N*C*H_2_ DBU), 29.2, 29.1, 26.8, 23.8, 20.6 (5C, 5 × *C*H_2_ DBU) ppm; UHR ESI-QTOF: m/z calcd for C_32_H_39_N_2_O_4_S [M]^+^ 547.2625; found: 547.2624.

*8-N-[phenyl 2-O-benzoyl-3-O-benzyl-6-deoxy-6-yl-1-thio-α-d-mannopyranoside]-1,8-diazabicyclo(5.4.0)undec-7-ene-chloride (**52**).* Compound **16 [27]** (50 mg, 0.058 mmol) was converted to **52** according to general **Method B**. The crude product was purified by silica gel chromatography (85:15 CH_2_Cl_2_/MeOH) to give **52** (32 mg, 76%) as a light yellow syrup. [α]_D_^25^+11.1 (*c* 0.56, CHCl_3_); *R*_f_ 0.41 (85:15 CH_2_Cl_2_/MeOH); ^1^H NMR (500 MHz, CDCl_3_) *δ* = 8.07–7.24 (m, 15H, arom.), 5.74 (s, 1H, H-1), 5.68 (s, 1H, H-2), 4.76 (d, *J* = 11.5 Hz, 1H, BnC*H*_2_a), 4.63 (d, *J* = 11.5 Hz, 1H, BnC*H*_2_b), 4.32 (t, *J* = 8.4 Hz, 1H, H-5), 4.14 (d, *J* = 15.3 Hz, 1H, H-6a), 3.97 (t, *J* = 9.3 Hz, 1H, H-4), 3.89 (dd, *J* = 2.6 Hz, *J* = 9.0 Hz, 1H, H-3), 3.83 (dd, *J* = 8.2 Hz, *J* = 15.5 Hz, 1H, H-6b), 3.57–3.42 (m, 6H, 3 × NC*H*_2_ DBU), 2.91 (dd, *J* = 6.6 Hz, *J* = 15.4 Hz, 1H, C*H*_2_a DBU), 2.76 (dd, *J* = 8.6 Hz, *J* = 14.4 Hz, 1H, C*H*_2_b DBU), 1.90–1.55 (m, 9H, H-4-O*H*, 4 × C*H*_2_ DBU) ppm; ^13^C NMR (125 MHz, CDCl_3_) *δ* = 167.9 (1C, C_q_ DBU), 165.8 (1C, C_q_ Bz), 137.5, 133.0 (3C, 3 × C_q_ arom.), 131.4–128.0 (15C, arom.), 85.4 (1C, C-1), 77.1 (1C, C-3), 72.7 (1C, C-5), 72.2 (1C, Bn*C*H_2_), 70.4 (1C, C-2), 68.5 (1C, C-4), 55.7 (1C, N*C*H_2_ DBU), 55.2 (1C, C-6), 49.5, 48.4 (2C, 2 × N*C*H_2_ DBU), 28.9, 28.5, 26.0, 23.0, 20.0 (5C, 5 × *C*H_2_ DBU) ppm; UHR ESI-QTOF: m/z calcd for C_35_H_41_N_2_O_5_S [M]^+^ 601.2731; found: 601.2729.

*8-N-[phenyl 3-O-benzyl-6-deoxy-6-yl-4-O-(2′-naphthyl)methyl-1-thio-α-d-mannopyranoside]-1,8-diazabicyclo(5.4.0)undec-7-ene-acetate (**53**).* Compound **16**^27^ (70 mg, 0.081 mmol) was converted to **53** according to general **Method A**. The crude product was purified by silica gel chromatography (85:15 CH_2_Cl_2_/MeOH) to give **53** (47 mg, 85%) as a light yellow syrup. [α]_D_^25^+106.7 (*c* 0.12, MeOH); *R*_f_ 0.57 (85:15 CH_2_Cl_2_/MeOH); ^1^H NMR (500 MHz, CDCl_3_) *δ* = 7.85–7.26 (m, 17H, arom.), 5.86 (s, 1H, H-1), 5.08–4.65 (m, 4H, NAPC*H*_2_, BnC*H*_2_), 4.36 (s, 1H, H-2), 4.12 (t, *J* = 8.4 Hz, 1H, H-5), 3.90 (dd, *J* = 3.0 Hz, *J* = 8.9 Hz, 1H, H-3), 3.84 (t, *J* = 9.3 Hz, 1H, H-4), 3.81–3.77 (m, 2H, H-6a,b), 3.53–3.28 (m, 6H, 3 × NC*H*_2_ DBU), 2.78–2.74 (m, 1H, C*H*_2_a DBU), 2.66–2.62 (m, 1H, C*H*_2_b DBU), 2.16 (s, 3H, C*H*_3_COO^−^), 1.69–1.46 (m, 9H, H-2-O*H*, 4 × C*H*_2_ DBU) ppm; ^13^C NMR (125 MHz, CDCl_3_) *δ* = 167.3 (1C, C_q_ DBU), 137.4, 135.1, 133.3, 133.1, 132.9 (5C, 5 × C_q_ arom.), 130.6–126.2 (17C, arom.), 86.0 (1C, C-1), 79.8 (1C, C-3), 75.5 (1C, C-4), 75.4 (1C, NAP*C*H_2_), 71.7 (1C, Bn*C*H_2_), 71.1 (1C, C-5), 68.5 (1C, C-2), 55.7 (1C, N*C*H_2_ DBU), 55.0 (1C, C-6), 49.4, 48.0 (2C, 2 × N*C*H_2_ DBU), 31.0 (1C, *C*H_3_COO^−^), 29.0, 28.3, 25.8, 22.7, 19.7 (5C, 5 × *C*H_2_ DBU) ppm; UHR ESI-QTOF: m/z calcd for C_39_H_45_N_2_O_4_S [M]^+^ 637.3095; found: 637.3095.

### 3.2. Antimicrobial Investigations of the DBU-Conjugated Derivatives

The detection of the antibacterial and antifungal effects of the tested compounds and the determination of the minimum inhibitory concentration (MIC) were carried out based on the method of CLSI [48] and Hungarian Pharmacopeia (chapters VIII. 2.6.12–13.) [49].

#### 3.2.1. Materials

For the antimicrobial studies, we used three human pathogens as test organisms: *Candida albicans* (ATCC 10231), *Staphylococcus aureus* subspecies *aureus* (ATCC 6538) and *Escherichia coli* (ATCC 8739).

*Staphylococcus aureus* subspecies *aureus* and *Escherichia coli* were cultured on tryptic soya agar (TSA) for 18–24 h at 37 °C. *Candida albicans* was cultured on Sabouraud dextrose agar (SDA) for 18–24 h at 37 °C. To obtain cells for suspension, we made cell suspensions from *Staph. aureus*, *E. coli* and *C. albicans* with the cotton swab suspension technique. The dilution of suspension was then made with 0.85% NaCl solution, absorbance being set to A_600nm_ = 0.06. a volume of 10 μL of these suspensions contained ~4.5–6.0 × 10^4^ colony forming unit (CFU) (bacteria) and ~1.5–2.0 × 10^3^ CFU for yeast.

Tested agents were diluted with DMSO; the concentration of stock solutions was 4 mg/mL; and to ensure sterility, the closed sample containers were incubated at 35 °C for 1 h. The highest final concentration was 100 μg/mL.

#### 3.2.2. Methods

Testing was performed in 24-well plates, and cell suspensions were prepared with 0.85% NaCl solution. Stock solutions were made from the agents with DMSO (100%). Final concentrations of used compounds were 6.25–100 µg/mL. The activity of the three most effective antibacterial (**10**, **12** and **22**) and antifungal (**10**, **22** and **29**) compounds was also determined at lower concentrations (10× diluted); the compounds were tested against *S. aureus* and *C. albicans* at the concentration range of 0.625–10.0 µg/mL. After inoculation, plates were incubated at 35 °C, and results were then read visually after 18–24 h. If microbial growth could not be detected, results were read visually after 72 h. If there was no growth after 72 h, incubation was continued till the fifth day of the experiment and the results were read visually and with plate reader at 600 nm after 120 h.

#### 3.2.3. Evaluation

The MIC value of a compound was defined as the minimal concentration that inhibits the growth of bacteria and yeasts comparing to the control. The results of the antibacterial and antifungal tests were read visually after 24, 48, 72 and 120 h. The microbial growth was measured by plate reader at 600 nm after 120 h.

The IC_50_ value of a compound was defined as the concentration that inhibits cell growth by 50% relative to the control. After the MTT test, the medium containing the inhibitor was removed, and 10 μL MTT solution was added to the cells. The plates were then incubated for two hours at 37 °C. The medium from the wells was then carefully aspirated, and MTT formazan dissolved with 100 μL of DMSO aided by gentle agitation on a shaker. After 10 min at room temperature, the absorbances were read at 570 nm by an automatic plate reader. The percentage of viability reflecting the respiratory potential of the cell population in each well was expressed as (absorbance of treated cells/absorbance of control cells) × 100. IC_50_ values of tested agents were determined by Graphpad (Prism) semi-log line fitting (graphpad.com (8)) [50]. The MTT test was performed after treatment and removal of inhibitors followed by the addition of 10 μL of MTT solution to the cells.

### 3.3. Viability Tests

We tested the toxic effect of compounds **3**, **10**, **12**, **21, 22** and **29** on HaCaT cell line. HaCaT is a spontaneously transformed, aneuploid keratinocyte, an adherent cell type derived from the epithelium of Human Caucasian adult male skin. The cell line was immortalized in 1986 under low calcium and high-temperature conditions [51]. Cells were cultured in DMEM medium (iBiotech, Szigetszentmiklós, Hungary) containing 10% fetal bovine serum (FBS, iBiotech, Szigetszentmiklós, Hungary) and 1% antibiotic-antimycotic mix (Penicillin–Streptomycin–Neomycin) [52].

### 3.4. Hemolytic Activity

Blood was collected from a Sprague-Dawley rat into an EDTA tube. Erythrocytes were separated from blood by centrifugation at 1800 rpm for 8 min, washed 3 times with phosphate buffered saline (PBS), and resuspended in PBS. Aliquots of the cell suspension with the respective red blood cell number of 7 × 10^7^ were added to PBS containing increasing concentrations of the test compounds or the vehicle DMSO. After gently mixing, each mixture was incubated at 37 °C for 15 min. The released hemoglobin into the supernatant was separated by a rapid centrifugation at 5000 rpm for 5 min. Finally, the absorbance of the released hemoglobin was measured at 540 nm using a Multiskan Go (ThermoFisher, Waltham, MA, USA) microplate reader. The absorbance values were compared to the absorbance of the sample kept in purified water during the incubation as positive control and given as percent hemolysis [53].

### 3.5. Membrane Permeabilization Study of Lactobacillus plantarum by SYTOX Green Staining

*Lactobacillus plantarum* subsp. *plantarum* (ATCC 14917) was purchased from ATCC (Manassas, VA, USA). The membrane permeabilization ability of compounds **10** and **22** was tested on *L. plantarum* and determined in the brain heart infusion medium (BHI) (CliniChem Ltd., Budapest, Hungary). Bacterial cells were treated with the test compounds at the final concentration of 6.25 µg/mL and samples were taken at 1, 4, 8, 24 and 48 h. Samples were stained with SYTOX green fluorescent dye (ThermoFischer Scientific, Budapest, Hungary) at 1 µM final concentration for 30 min at 37 °C, and flow cytometric measurements were carried out with a Guava Easy Cyte 6HT-2L flow cytometer (Merck Ltd., Darmstadt, Germany). Green (525/30 nm) and red (695/50 nm) fluorescence channels were used to gate out the cells on a green versus red dot plot. The ratio of the SYTOX green (SG) positive bacterial cells was evaluated [54].

## 4. Conclusions

In the course of our research, we produced novel DBU-conjugated cationic carbohydrate derivatives from d-hexopyranosides of various configurations (d-*gluco*, d-*manno*, d-*allo* and d-*altro*). The -OH groups of carbohydrates were masked with benzyl, benzoyl and/or 2-naphthylmethyl protecting groups. By varying these protecting groups and free hydroxyls on the sugar unit, we were able to tune the lipophilicity of the molecules. A total of 13 fully protected and 9 partially or completely deprotected cationic sugar amidinium salts were prepared. We optimized the synthesis of a d-*gluco*-DBU conjugate, and attempted to produce carbohydrate amidinium derivatives without the use of protecting groups.

The conjugation of the DBU unit to the sugar skeleton was carried out by nucleophilic substitution reactions on the fully protected 6-deoxy-6-iodohexopyranosides. The efficiency of the desired nucleophilic substitution was strongly dependent on the configuration of the sugar used due to competitive elimination reaction. We have found that the more the C-5 axial hydrogen of the carbohydrate is sterically hindered, the more favored the nucleophilic substitution reaction is. Accordingly, in the case of d-*altro* and d-*allo* configurations, where H-5 is sterically shielded, nucleophilic substitution was preferred, leading to the expected DBU-conjugated derivative in good yield. At the same time, in the case of the d-*gluco* and d-*manno* configurations, due to the good steric accessibility of H-5, elimination was the dominant reaction, and unfortunately the optimization experiments resulted in only a small, 7% increase in the yield of the DBU-conjugated glucoside.

Our attempts to prepare unprotected DBU-sugar conjugates without the use of protecting groups failed. In the presence of free hydroxyls, intramolecular nucleophilic substitution took place in all investigated cases, resulting in the 3,6-anhydro derivative of the carbohydrate.

Of the twenty-two compounds tested, nine showed promising activity against *Staphylococcus aureus* with MIC values of 6–25 µg/mL, and six compounds showed potent antifungal activity against *Candida albicans*. The biological results clearly show that protecting groups are necessary for antifungal and antibacterial activity and that ether protecting groups, especially naphthylmethyl ether, are more favorable than ester protecting groups. The most effective compounds turned out to be the most lipophilic derivatives of highest clogP containing only ether protecting groups. Presumably, the lipophilic units facilitate the incorporation of the compounds into the lipid bilayer of the cell membrane and disrupt it. The positive results of membrane permeabilization experiments performed on a Gram-positive model bacterium with two active compounds (**10** and **22**) confirmed that the antimicrobial effect of our DBU-sugar conjugates may be related to their membrane-damaging ability. With the appearance of free hydroxyl groups, biological activity dropped drastically, and molecules containing more than one free hydroxyl group completely lost their activity. At the same time, the sugar configuration also plays a role in the effect, as evidenced by the ineffectiveness of the *altro*-configured compounds.

We found that the pattern of the protecting group also affects the cytotoxic properties. Compounds containing an ester group on the molecule or containing a benzyl group in position 4 (**3**, **12** and **21**) were cytotoxic on HaCaT cells. However, derivatives with exclusively ether groups on the molecule and a NAP group at position 4 (**10**, **22** and **29**) showed a greatly reduced toxicity. The latter three compounds had the best antifungal effect. They act against fungi at concentrations lower than their cytotoxicity value, thus giving the opportunity to use these derivatives as antifungal agents in either oral or dermal antifungal preparations. However, this still requires further investigations.

None of the 22 tested compounds had an effect against the Gram-negative *Escherichia coli* (ATCC 8739) strain in the applied concentration range of 6.25–100 µg/mL. Considering the cationic, membrane-active structure of our compounds, this is a surprising result and worthy of further investigation.

## Data Availability

Not applicable.

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
