# Peer review of "First Synthesis of DBU-Conjugated Cationic Carbohydrate Derivatives and Investigation of Their Antibacterial and Antifungal Activity"

_ijms, 2023, doi:10.3390/ijms24043550_

Round 1

Reviewer 1 Report

According to the advantage of cationic compounds, the authors designed and synthesized a series of DBU-derived amidinium salts of carbohydrates and investigated their antibacterial and antifungal Activity. A series of glucose-DBU conjugates were synthesized by optimizing the reaction conditions and confirmed by NMR and HRMS. The effect of the obtained quaternary amidinium salts against Escherichia coli and Staphylococcus aureus bacterial strains and Candida albicans yeast was investigated, and the impact of the used protecting groups and the sugar configuration on the antimicrobial activity was analyzed. This work has highly innovative and the methods are described adequately, so it can be accepted for publication in this journal. However, there are still two issues to be solved or explained.

1. The authors predicted that these compounds may be suitable for disturbing the cell walls of bacteria and fungi due to their quaternary ammonium moiety. Whether corresponding experiments have been carried out to verify it.

2. In the antibacterial and antifungal activity test, it is recommended to add the control drugs or quote the results of control drugs.

Author Response

A point-by-point response to the Reviewer comments:

Reviewer 1:

  1. “The authors predicted that these compounds may be suitable for disturbing the cell walls of bacteria and fungi due to their quaternary ammonium moiety. Whether corresponding experiments have been carried out to verify it.”

Answer: There were no such tests in the first submitted paper, however, at the reviewer's suggestion, in the case of two compounds (10 and 22) that were effective against both fungi and Gram-positive bacteria, we performed the membrane permeabilization experiments against Gram-positive Lactobacillus plantarum and thus supplemented the biological investigation part of the revised manuscript (page 9-10, Figure 5.). These experiments verified the membrane disturbing ability of our compounds. Accordingly we inserted the following sentence in the Conclusion section: “The positive results of membrane permeabilization experiments performed on a Gram-positive model bacterium with two active compounds (10 and 22) confirmed that the antimicrobial effect our DBU-sugar conjugates may be related to their membrane-damaging ability.”

  1. “In the antibacterial and antifungal activity test, it is recommended to add the control drugs or quote the results of control drugs.”

Answer: Thank you for this suggestion. We have added the activity of the reference drugs in Table 2. We chose Ceftazidime and Gentamicin as antibacterial reference compounds, and Amphotericin B as an antifungal reference compound.

Reviewer 2 Report

The manuscript is written well but falls short on a few aspects. I would recommend the authors to optimize further. Please see my comments below.

Critical:

1.     The chemistry is not novel, and the compound class has been reported earlier.

2.     The biological activity is also not remarkable and is extremely limited.

3.     It is more common to find compounds that are active against gram positive bacteria than gram negative bacteria. The expectation/premise of this class of compounds is the activity against gram negative bacteria, but the authors failed to explain the reason why such activity is not observed.

4.     A minimum of four-fold difference in concentration is expected between the IC50 of cytotoxicity and MIC and it is barely two-fold in this case.

5.     Why was hemolytic activity not measured as part of the cytotoxicity studies?

Miscellaneous:

1.     More references can be added in the third paragraph for the mechanism of action of “positively charged compounds”.

2.     NMR spectra are not very clean, and the integration is shady and clumsy.

3.     Reference formatting is not consistent and must be uniform.

4.     Table 2 has typographical errors.

5.     NMR write-up for some compounds also has typographical errors.

Author Response

A point-by-point response to the Reviewers’ comments:

Reviewer 2:

  1. “The chemistry is not novel, and the compound class has been reported earlier.”

Answer: The reviewer is right, the chemical reactions by which our compounds were prepared are indeed known. However, we were the first to isolate and identify sugar-DBU-conjugates, and then optimize these reactions for the production of DBU-conjugated amidinium derivatives. In the literature, there are indeed compounds conjugated with similar bases (e.g. DABCO), but no one has so far produced DBU-linked derivatives or investigated the biological effects of these compounds.

  1. “The biological activity is also not remarkable and is extremely limited.”

Answer: Regarding the biological activities, unfortunately, before testing, we can only hypothesize the effect of the synthesized compounds. Any synthetic chemist would be happy to know in advance that the molecule he/she has designed and prepared will have an excellent effect. Many times, even derivatives designed by in silico calculations and predicted to be highly active turn out to be of low efficiency. Indeed, only some of our compounds have antimicrobial effects but among the active derivatives we have identified some promising structures with relatively good biological activity.

  1. “It is more common to find compounds that are active against gram positive bacteria than gram negative bacteria. The expectation/premise of this class of compounds is the activity against gram negative bacteria, but the authors failed to explain the reason why such activity is not observed.”

Answer: Indeed, activity against Gram-negative bacteria can be assumed for quaternary ammonium compounds. Unfortunately, our compounds did not prove to be active, which is certainly surprising. One explanation for this inactivity could be that our compounds do not interact properly with the lipopolysaccharides embedded in the outer membrane of Gram-negative bacteria. But this is just an assumption, and proving it would require a lot of work, the outcome of which is uncertain.  However, since some of our compounds showed promising antifungal activity, we will focus on the optimization of these structures in the future.

  1. “A minimum of four-fold difference in concentration is expected between the IC50 of cytotoxicity and MIC and it is barely two-fold in this case.”

Answer: Indeed, the condition for safe use is a fourfold IC50/MIC difference. In our case, however, we presented the first results of basic research that had just started, in the framework of which we studied the antimicrobial effect of sugar-DBU conjugates for the first time. At the same time, we investigated which compounds could be effective and to what extent. In light of these results, as a continuation of the research, we may have the opportunity to design/synthesize even more effective and less toxic derivatives.

  1. Why was hemolytic activity not measured as part of the cytotoxicity studies?

Answer: At the reviewer's suggestion, in the case of two compounds (compounds 10 and 22), which were effective against both fungi and Gram-positive bacteria, we performed the suggested hemolytic activity test and supplemented the manuscript with their results (page 9, Figure 4.).

  1. “More references can be added in the third paragraph for the mechanism of action of “positively charged compounds”

Answer: The literature references of the mechanism paragraph have been supplemented with additional references as requested (references 24, 25 and 26 in the revised version).

  1. “NMR spectra are not very clean, and the integration is shady and clumsy.”

Answer: Due to the highly ionic nature of our compounds, the more sensitive 1H NMR spectra may show signs of increased multiplicity. In all cases, we tried to create the best possible quality spectra by changing the parameters. Despite the fact that the compounds were always converted into a uniform salt with an ion exchange resin, it was usually observed that they replaced their existing counterion as soon as the opportunity arose. The "shady" nature of the spectra may result from the presence of this ion mixture. However, it can be seen from the 13C spectra that our compounds were free of impurities.

  1. “Reference formatting is not consistent and must be uniform.”

Answer: We checked and standardized the references in accordance with the formal requirements.

  1. “Table 2 has typographical errors.”

Answer: Table 2 has been typographically standardized.

  1. “NMR write-up for some compounds also has typographical errors.”

Answer: The typing and typographical errors have been corrected in the experimental section.

Round 2

Reviewer 2 Report

I am satisfied with the responses of authors and the additions/alterations made to the manuscript.